# Advancement in Clinical Glycomics and Glycoproteomics for Congenital Disorders of Glycosylation: Progress and Challenges Ahead

**DOI:** 10.3390/biomedicines13081964

**Published:** 2025-08-13

**Authors:** Nurulamin Abu Bakar, Nurul Izzati Hamzan

**Affiliations:** 1Centre of Diagnostics, Therapeutics and Investigative Studies (CODTIS), Faculty of Health Sciences, Universiti Kebangsaan Malaysia, Jalan Raja Muda Abdul Aziz, Kuala Lumpur 50300, Malaysia; 2Special Protein Unit, Specialized Diagnostic Centre, Institute for Medical Research, National Institutes of Health, Jalan Pahang, Kuala Lumpur 50588, Malaysia; nurulizzati.h@moh.gov.my

**Keywords:** clinical glycomics, glycoproteomics, congenital disorders of glycosylation, mass spectrometry, biomarker discovery, multi-omics integration

## Abstract

Congenital disorders of glycosylation (CDG) are a group of rare, multisystemic genetic diseases caused by defects in glycan biosynthesis and protein glycosylation. Their broad clinical and genetic heterogeneity often require advanced diagnostic strategies. Clinical glycomics and glycoproteomics have emerged as powerful tools for understanding and diagnosing CDG by enabling high-resolution analysis of glycan structures and glycoproteins. Advancements in high-throughput mass spectrometry (MS) and site-specific glycoproteomics have led to the identification of disease-relevant biomarkers, providing insight into underlying glycosylation defects. These technologies enable detailed analysis of glycan structures and glycoproteins, improving early diagnosis, supporting biomarker discovery, and facilitating therapy monitoring. Integration with genomic and clinical data, including the use of dried blood spot testing and isotopic tracing, further enhances diagnostic precision and reveals the functional consequences of pathogenic variants. While challenges remain in standardizing methods, ensuring accessibility, and implementing bioinformatics tools, global collaborations and harmonized guidelines are beginning to address these gaps. Future directions include the use of artificial intelligence in data analysis, the development of comprehensive diagnostic frameworks, and international efforts to standardize glycomic methods. Collectively, these advances reinforce the growing clinical value of glycomics and glycoproteomics in the diagnosis and management of CDG.

## 1. Introduction

Congenital disorders of glycosylation (CDG) are a group of rare, multisystemic genetic conditions caused by defects in glycosylation pathways. Each CDG subtype is named after the causative gene, followed by the suffix -CDG, reflecting the underlying molecular defect. For example, the most prevalent type of CDG, phosphomannomutase 2 deficiency, is abbreviated as PMM2-CDG [1]. With over 160 different CDG types currently identified, these disorders exhibit broad clinical and genetic heterogeneity [2]. Clinical glycomics refers to the application of glycan analysis to investigate disease-related glycosylation changes for diagnostic, prognostic, and therapeutic purposes. It translates complex glycan profiles into clinically actionable insights, particularly in CDG, where glycosylation defects are central to disease pathophysiology. By utilizing advanced mass spectrometry (MS) technologies, clinical glycomics can deliver deep insights into patient-specific glycan profiles, enabling earlier diagnosis and more targeted therapeutic strategies. Two major analytical focuses include N-glycan profiling, which can be performed globally through total serum or plasma N-glycan analysis, or more specifically by analyzing transferrin glycosylation, a well-established marker for the majority of CDG types, particularly those affecting the N-glycosylation pathway. In contrast, O-glycan analysis, often targeting apolipoprotein C-III (ApoCIII), provides diagnostic information for mucin-type glycosylation defects, such as those observed in subtypes like COG6-CDG and B4GALT1-CDG, where abnormalities in O-glycosylation have been reported [3]. In addition, dried blood spot (DBS) testing has emerged as a practical, cost-effective, and minimally invasive approach for sample collection and transport, offering significant advantages for early diagnosis and newborn screening in rare genetic disorders [4].

Combined genetic testing, such as the use of next-generation sequencing technology alongside glycomic profiling, enhances diagnostic precision and facilitates resolution of uncertain genetic variants. In recent years, whole-exome sequencing (WES) has emerged as a valuable tool for diagnosing CDG, especially in patients with unclear or atypical clinical presentations. It has successfully identified pathogenic mutations in genes associated with glycosylation and helped resolve previously undiagnosed cases [5,6,7]. However, while WES provides detailed genetic information, it does not offer insight into the functional consequences of these mutations at the glycan level. For instance, in phosphoglucomutase 1 deficiency (PGM1-CDG), glycomics revealed abnormal transferrin glycoforms and total plasma N-glycan profiles, which confirmed pathogenicity and guided therapeutic decisions, as well as offering additional insight into the structural defects in glycosylation. It also helped speed up the diagnosis. Therefore, biochemical characterization through glycomics and glycoproteomics remains crucial for confirming pathogenicity and for understanding structural defects in glycosylation [8].

The clinical heterogeneity of CDG requires a comprehensive diagnostic strategy. Biochemical screening using glycomics has been shown to expedite diagnosis and improve prognosis, as demonstrated in PGM1-CDG using intact transferrin and total plasma glycoprofiling [8]. In parallel, advances in glycoproteomics, particularly in site-specific glycosylation profiling, have linked specific genetic mutations to altered glycan structures and associated phenotypes [9]. Furthermore, consensus guidelines, such as those established for PGM1 CDG, have supported the harmonization of diagnostic and therapeutic approaches, thereby enhancing consistency in patient management [10].

The integration of glycomics with other omics technologies, such as genomics, transcriptomics, and metabolomics, has expanded our understanding of CDG pathophysiology. This review provides an updated overview of the key advancements in clinical glycomics and glycoproteomics, with emphasis on the emerging biomarkers, multi-omics integration, and diagnostic strategies that are shaping the future of clinical practice in CDG.

## 2. Technological Advancements

Over the past decade, continuous innovations in MS technologies have significantly advanced the diagnostic capabilities for CDG, particularly in resolving complex glycan structures. High-sensitivity methods such as porous graphitized carbon (PGC) liquid chromatography (LC) MS (PGC-LC-MS) and quadrupole time-of-flight (QTOF) MS (QTOF-MS) have provided researchers and clinicians with enhanced capabilities to perform detailed glycan and glycoprotein profiling, including the structural resolution of isomeric forms [11,12]. In particular, the adoption of soft ionization methods, such as electrospray ionization (ESI), for intact transferrin glycoprofiling has improved structural resolution and diagnostic accuracy [13,14]. A targeted LC tandem mass spectrometry (MS/MS) (LC-MS/MS) approach using DBS has also been developed for intact transferrin glycoform analysis, offering a high-throughput and clinically robust alternative to traditional isoelectric focusing methods for CDG diagnosis and newborn screening [15].

Automation with high throughput workflows has streamlined glycomic diagnostics by reducing the manual processing time and improving reproducibility. This aligns with recent technical standards established by the American College of Medical Genetics and Genomics (ACMG), which emphasize the implementation of validated workflows for transferrin glycoform analysis and broader biochemical testing in CDG [16]. Advances in sample preparation, such as solid-phase extraction, glycan derivatization, and glycan labeling, have further increased detection sensitivity and reliability. These methods combined with high-throughput analytical platforms have enabled large-scale glycomic studies in clinical and research settings and improved both reproducibility and diagnostic utility in glycosylation disorders [17]. Matrix-assisted laser desorption ionization time-of-flight MS (MALDI TOF MS) remains a valuable tool for rapid screening; it is particularly useful in resource-limited settings or for high-throughput population studies due to its minimal sample preparation and cost efficiency [18].

Significant strides have also been made in the analysis of O-glycans. For example, PGC-nanoLC-MS now enables high-resolution profiling of O-glycan structures without derivatization, eliminating the need for labor-intensive chemical labeling and simplifying analysis [19]. ApoCIII MS profiling remains a practical diagnostic marker for mucin-type O-glycosylation defects, such as those observed in COG6-CDG, while modern LC-MS/MS platforms facilitate concurrent quantification of multiple apolipoproteins, including ApoCIII and others relevant to glycosylation status, supporting broader clinical biomarker panels [20,21].

## 3. Integration with Multi-Omics

Recent advances have highlighted the value of integrating glycomics and glycoproteomics with other omics platforms, including transcriptomics, proteomics, and metabolomics, to enhance our understanding of CDG mechanisms and improve clinical interpretation. These integrative approaches have been applied in disease models to investigate glycosylation remodeling at the systems level, revealing unique glyco-transcriptomic signatures, protein glycosylation changes, and metabolic shifts associated with disease progression and therapy response. Such studies have identified novel glycoproteins and glycan alterations that contribute to cellular dysfunction and resistance mechanisms, offering insights that may be translatable to CDG research and biomarker development [22,23].

For example, in SRD5A3-CDG, glycomics alone may yield overlapping or inconclusive glycan profiles. In such cases, integration with transcriptomics or glycoproteomics can clarify the impact of pathogenic variants by linking genotype to functional glycosylation defects [24]. Alvarez and colleagues demonstrated that combining RNA-sequencing-based transcriptomics with N-glycomics enables the prediction of glycan biosynthesis and tissue-specific glycosylation patterns [25]. This strategy helps prioritize disease-related genes and interpret ambiguous variant effects. In addition, Van der Burgt and Wuhrer emphasized the role of multi-omics frameworks in precision diagnostics, underscoring how glycoproteomics integrated with genomics enhances diagnostic yield in rare glycosylation disorders [26].

Although historically less emphasized, metabolomics is now increasingly recognized as a complementary tool in CDG research. Targeted metabolomic profiling of patient samples has been applied to map disease-specific metabolic signatures, such as in PGM1-CDG and PMM2-CDG, offering insight into underlying biochemical disturbances [27,28]. These strategies support variant interpretation and therapy monitoring. Isotopic tracing has gained attention as a mechanistic tool for investigating glycosylation pathways. By following the incorporation of labeled sugars into nucleotide sugar pools and glycan structures, this method provides valuable insights into functional glycosylation defects and supports the biochemical validation of candidate variants. In the context of CDG, isotopic tracing offers a promising approach to elucidate metabolic bottlenecks and assess the efficiency of glycosylation in cellular systems [29].

In addition, metabolomics has revealed broader systemic effects in CDG and can be particularly informative in cases with combined metabolic and glycosylation defects. A recent study demonstrated complex metabolomic signatures in a patient with both PGM1-CDG and mitochondrial dysfunction, reflecting altered glycan synthesis and energy metabolism [30]. When integrated with glycoproteomics, metabolomic data enhances the interpretation of biochemical phenotypes and supports variant classification [31,32].

Systems biology approaches integrating glycomics with transcriptomics, proteomics, and metabolomics offer a powerful means to reconstruct disease pathways and guide individualized therapy [33]. For example, transcriptomic data combined with N-glycomics has been used to predict glycan biosynthesis and tissue-specific expression patterns, offering a basis for structure-informed diagnostics [25]. This has been demonstrated in studies of ALG1-CDG and PGM1-CDG. These strategies can be applied to CDG to identify affected biosynthetic pathways, classify overlapping phenotypes, and support precision diagnostics, particularly where genetic findings are inconclusive or phenotypic overlap complicates diagnosis.

In recent years, glycoproteomics has emerged as a powerful approach to detect subtle, site-specific glycosylation abnormalities in CDG, facilitating both diagnosis and mechanistic understanding of affected pathways. Glycoproteomics provides site-specific glycosylation data that complements metabolomics by enabling deeper interpretation of disrupted pathways and supporting therapy monitoring. This integration strengthens personalized diagnostics, particularly in CDG patients receiving dietary or enzyme replacement therapies, where glycan remodeling is treatment responsive [26]. Recent studies have extended the application of glycoproteomics to both patient-derived cells and clinical biofluids. In ALG1-CDG fibroblasts, glycoproteomic profiling has revealed a dysregulated glycoproteome that supports both diagnostic refinement and insight into disease mechanisms at the cellular level [34]. Similarly, in SRD5A3-CDG fibroblasts, combined glycoproteomic and proteomic analysis revealed distinct N-glycosylation defects that affect protein function, localization, and stability [24]. Glycoproteomic profiling of cerebrospinal fluid has uncovered brain-specific glycosylation changes that may help explain the neurological features observed in certain CDG subtypes and offer potential as fluid-based biomarkers to monitor neurological involvement in CDG [35].

Figure 1 presents a clinical and exploration multi-omics workflow for PGM1-CDG, showing how glycomics, glycoproteomics, and metabolomics contribute from diagnosis to biomarker discovery. This case highlights how integrative omics approaches applied to well-characterized subtypes like PGM1 CDG can link clinical evaluation to exploratory biomarker discovery and therapy monitoring. This framework captures both current diagnostic practice and the future potential of research-driven biomarker development in CDG.

The figure illustrates an integrated clinical and research workflow for PGM1-CDG. The process begins with clinical assessment based on characteristic features such as hepatopathy, hypoglycemia, and growth delay. This is followed by sample collection for transferrin glycoprofiling and N-glycan analysis using platforms such as QTOF-MS and LC-MS/MS. Genetic confirmation through PGM1 sequencing establishes the molecular diagnosis. In parallel, research-level integration of glycoproteomics and metabolomics facilitates the exploration of biomarker discovery. This comprehensive strategy not only enhances diagnostic precision and therapy monitoring but also paves the way for novel biomarker identification and individualized care in CDG.

## 4. Biomarker Discovery

The identification of solute carrier family 10, member 7 (SLC10A7) as a regulator of bone development and Golgi glycosylation demonstrates how integrating glycomics with genomics can reveal gene–phenotype relationships and elucidate disease mechanisms [36]. Similarly, the use of single-molecule molecular inversion probes combined with glycomics has improved the classification of CDG type I subtypes, such as PMM2-CDG and ALG1-CDG. PMM2-CDG, being the most prevalent subtype, benefits from early glycomic markers for timely diagnosis. Furthermore, ALG1-CDG presents diagnostic challenges due to mild biochemical phenotypes and uncertain variant interpretation, where glycomic analysis helps resolve functional ambiguity [37].

Table 1 summarizes the glycomics-derived biomarkers (glycomarkers) reported in selected CDG subtypes, including PGM1-CDG, SLC10A7-CDG, PMM2-CDG, and ALG1-CDG. These glycomarkers have demonstrated diagnostic utility and show potential for application in therapy monitoring.

Quantitative glycomics has also enabled the discovery of subtype-specific markers, such as a diagnostic tetrasaccharide in MOGS-CDG, expanding the clinical spectrum and diagnostic reach [41]. The identification of this glycomarker by high-throughput profiling not only aids early diagnosis but also broadens our understanding of glycosylation defects in CDG subtypes beyond the most common N-glycan profiles. This marker demonstrates how quantitative glycomics can uncover disease-specific glycan signatures that complement genetic and enzymatic testing in challenging diagnostic cases.

Building upon traditional glycomic biomarkers, glycoproteomics now allows for more refined, protein- and site-specific insights, which is essential for resolving biochemical phenotypes in complex CDG presentations. Recent studies have strengthened the position of glycoproteomics as a powerful platform for biomarker discovery in CDG. Plasma glycoproteomics has demonstrated high specificity in detecting site-specific glycosylation abnormalities that distinguish CDG from other metabolic disorders, supporting its utility for differential diagnosis and patient stratification [42]. The ability of glycoproteomics to quantify site-specific glycosylation has been demonstrated in CDG patient cells, where unique glycan abnormalities were linked to underlying genetic defects [34]. Moreover, recent work has identified a complement C4-derived glycopeptide as a diagnostic biomarker for PMM2-CDG, further establishing the clinical relevance of glycoproteomics in biomarker development [43].

The profiling of isomeric glycan structures adds an additional layer of diagnostic resolution. For example, alpha 2,3-linked sialylated isomers have been associated with immune-mediated disorders such as Behcet’s disease, providing a model for how linkage-specific glycan analysis may benefit CDG diagnostics [44]. Complementary to this, high-sensitivity capillary electrophoresis and hydrophilic interaction chromatography with ultra-performance liquid chromatography (HILIC-UPLC) MS platforms now enable detailed mapping and high-throughput separation of glycan isomers. The application of HILIC-UPLC-MS for analyzing isomeric N-glycans in CDG patient samples has confirmed its diagnostic utility in resolving complex glycan structures relevant to glycosylation disorders [45]. In addition, LC-MS/MS has been used to profile isomeric N-glycans derived from low-abundance serum glycoproteins, demonstrating the sensitivity of this method in detecting subtle glycan variations that may support early disease detection or patient stratification [46].

In addition to diagnosis, glycoproteomic biomarkers serve as tools for therapy monitoring. Longitudinal glycan profiling in patients with PGM1-CDG receiving galactose supplementation has shown measurable glycosylation changes correlating with clinical improvement [47]. These markers function as surrogate endpoints and help guide treatment decisions over time. The adoption of high-throughput LC-MS/MS workflows allows for consistent and reproducible biomarker quantification across larger patient cohorts, bridging the gap between research discoveries and clinical implementation. Together, these advancements further establish glycoproteomics as a key platform in personalized diagnostics and therapeutic monitoring in CDG.

## 5. Standardization of Diagnostic Practices

International consensus guidelines have played a pivotal role in unifying the diagnostic practices in CDG. For instance, the PGM1-CDG guideline emphasizes a multi-pronged approach that integrates clinical features, biochemical analyses, and genetic testing to achieve accurate diagnosis. It also provides detailed guidance on long-term follow-up, including the monitoring of glycosylation markers, liver and endocrine function, and nutritional status. These recommendations help standardize patient care and support early therapeutic intervention, particularly in response to galactose supplementation [10].

Among the most reliable tools is intact transferrin glycoprofiling using high-resolution MS, which supports both diagnosis and longitudinal therapy monitoring, as shown in phosphoglucomutase 1 deficiency [6]. In addition, serum glycoprotein profiling allows the tracking of site-specific glycosylation changes after treatment, offering insights into how therapies influence glycan processing and trafficking. A transferrin-based treatment index has also been proposed to evaluate the response to D-galactose supplementation in PGM1-CDG, showing how glycosylation changes can serve as surrogate markers of therapeutic efficacy [47].

To promote diagnostic accessibility, especially in resource-limited settings, non-invasive sampling methods such as DBS for transferrin and ApoCIII profiling are increasingly used. These methods allow stable storage and easy transport, enabling remote testing and earlier detection of glycosylation defects [14]. As a result, they are now widely adopted in clinical workflows to support both diagnosis and therapy monitoring.

Recent technical standards from the ACMG have emphasized the need for validated laboratory protocols in CDG diagnostics. These protocols include quality-assured transferrin glycoform analysis, result interpretation, and standardized reporting aimed at harmonizing laboratory practices [16]. As new therapies emerge, MS-based platforms, including data-independent acquisition (DIA) methods, offer scalable and reproducible solutions for analyzing glycosylation changes over time. These tools support consistent measurements across patient cohorts and enable high-throughput monitoring of the therapeutic response [48].

Together, these developments highlight the close relationship between diagnostic standardization and therapeutic monitoring in CDG, underscoring the importance of integrating glycomics-based strategies into routine clinical care. Such integration is essential for ensuring diagnostic accuracy and equitable patient care worldwide.

Standardization of mass spectrometry workflows is essential to ensure consistent and accurate glycan analysis in clinical diagnostics. Validated protocols for sample preparation, instrument settings, data acquisition, and interpretation help reduce variability between laboratories. In clinical glycomics, established methods such as MALDI TOF for transferrin profiling and PGC-LC-MS for total plasma N glycome analysis have contributed to improved diagnostic reliability. These approaches rely on clearly defined performance criteria, including detection thresholds, retention time consistency, and reproducible identification of glycan structures. Ongoing collaboration through external quality assessment programs and interlaboratory comparisons continues to strengthen the analytical robustness of glycomics workflows and facilitates their integration into clinical diagnostic pathways. In addition, curated MS databases containing reference glycan spectra support standardized interpretation and facilitate cross-laboratory comparisons. These platforms improve diagnostic consistency and promote reproducibility in glycomics-based testing.

Efforts to standardize the diagnostic practices in CDG have gained momentum through international collaborations and quality assurance programs. Proficiency testing schemes, such as those coordinated by the European Research Network for Evaluation and Improvement of Screening, Diagnosis and Treatment of Inherited Disorders of Metabolism (ERNDIM), have significantly contributed to improving analytical reliability and harmonizing laboratory performance in diagnosing CDG and related disorders [49]. In parallel, expert consensus statements and diagnostic algorithms are being continuously refined to guide clinical interpretation across heterogeneous CDG presentations.

Networking platforms such as the European Reference Network for Hereditary Metabolic Disorders (MetabERN) serve as effective tools for knowledge dissemination, promoting collaboration among clinical and research communities and supporting the implementation of standardized diagnostic approaches across countries. These efforts ultimately enhance diagnostic accessibility and aim to improve the quality of life for patients and families affected by CDG [50].

In addition, Table 2 provides a comparative overview of selected CDG subtypes, summarizing their clinical characteristics, glycomarkers, and diagnostic tools. This table is intended to support harmonization by illustrating the diagnostic variability and common analytical strategies used across representative CDG presentations.

## 6. Challenges in Clinical Translation

Despite significant progress, several challenges remain in translating glycomics and glycoproteomics into routine clinical diagnostics for CDG. A key barrier is the limited access to advanced MS platforms and skilled personnel, including bioinformatics specialists, particularly in low-resource settings. The high costs associated with instrumentation, reagents, and bioinformatics infrastructure hinder widespread clinical adoption.

Another major issue is the lack of standardized protocols across laboratories. Differences in sample handling, glycan derivatization, data acquisition, and interpretation often result in inconsistent findings and reduced reproducibility. As emphasized in a recent study, harmonizing workflows and developing certified reference materials, such as standardized transferrin glycoform controls are critical for transforming glycomics from a research-intensive approach to a validated clinical diagnostic platform [31].

Informatics integration presents additional complexity. Clinical glycomics generates high-dimensional datasets that require advanced analytical pipelines for accurate interpretation due to glycan isomerism, structural branching, and linkage variability. The integration of glycomics with genomic, proteomic, and clinical metadata is essential but remains technically demanding. A perspective by Van der Burgt and Wuhrer emphasized the need for interoperable data standards, shared ontologies, and cross-disciplinary collaboration to effectively incorporate glycoproteomic data into precision medicine frameworks [26].

Moreover, translating multi-omics tools into clinical diagnostics requires thoughtful validation and clinical-grade implementation. A review by Hertzog and colleagues emphasized that integrating emerging omics platforms, including metabolomics and glycoproteomics, into clinical practice for inborn errors of metabolism requires not only technical readiness but also support across regulatory, educational, and infrastructural domains. Their review underscores the importance of multidisciplinary cooperation to enable clinical laboratories to adopt advanced -omics tools for diagnostic and therapeutic purposes [51].

Collectively, these challenges highlight the importance of coordinated efforts among researchers, clinicians, and policy makers to ensure that the benefits of glycomics and glycoproteomics reach patients through validated, scalable, and equitable clinical applications.

## 7. Role of Glycan Databases and Bioinformatics

The interpretation of complex glycomic and glycoproteomic data in CDG heavily depends on specialized bioinformatics platforms and curated databases. These tools support accurate glycan annotation, data integration, and structure–function correlation, thereby enhancing both biomarker discovery and clinical diagnostics.

UniCarbKB serves as a central glycoproteomics knowledge platform, consolidating experimentally determined glycan structures and glycoprotein data and enabling consistent annotation and searchability across studies [52]. GlyConnect, with its interactive analytical interface, facilitates the exploration of glycoprotein–glycan relationships, helping researchers map glycosylation patterns to disease phenotypes [53]. Meanwhile, GlyTouCan functions as a global glycan structure repository that assigns unique identifiers to glycan compositions, promoting data interoperability and international collaboration [54].

In addition, platforms such as GlycoWorkbench provide computational tools for the MS-based annotation and structural elucidation of glycans, streamlining data interpretation in clinical and research laboratories [55]. Collectively, these platforms enhance diagnostic precision and increase analytical throughput by offering curated resources and automated pipelines for structure annotation, pathway mapping, and visualization.

From a broader perspective, the field of glycoinformatics has matured into a core component of glycoscience, integrating data from genomics, proteomics, and clinical phenotypes. The application of glyco-bioinformatics is crucial to support functional interpretation of glycan-related changes and enable discovery-driven diagnostics. Recent perspectives highlight the importance of data standardization, computational interoperability, and multi-omics integration in advancing clinical glycoscience [56].

Furthermore, as noted by Packer and colleagues, the development and application of bioinformatics in glycomics have opened new avenues for biomarker discovery by linking structural glycan data with disease mechanisms [57]. These informatics tools not only support the diagnosis of known CDG subtypes but also the discovery of novel variants. By mapping observed glycan abnormalities to biosynthetic pathways, researchers can prioritize candidate genes and assess the pathogenic relevance of uncertain variants.

In clinical practice, gene prioritization for CDG diagnosis is often guided by the patient’s biochemical glycosylation profile and clinical phenotype. For example, a type I transferrin glycosylation pattern typically indicates defects in the early steps of N-glycan assembly, directing attention toward the genes involved in the cytosolic and endoplasmic reticulum-based synthesis and transfer of dolichol-linked oligosaccharides, such as PMM2-CDG and ALG1-CDG. Conversely, a type II pattern, reflecting abnormal glycan processing or trafficking, suggests defects in Golgi-associated glycosylation, implicating genes such as COG6-CDG or B4GALT1-CDG.

Additional gene filtering can be guided by phenotype-driven approaches, where features such as predominant liver involvement, developmental delay, or muscular symptoms suggest specific molecular pathways. Diagnostic gene panels and broader sequencing methods, including molecular inversion probes and whole-exome sequencing, are increasingly interpreted in combination with glycomic profiles to improve diagnostic precision. Public resources that consolidate gene–phenotype associations and glycosylation pathway data, such as Orphanet (https://www.orpha.net/en/disease, accessed on 9 July 2025) and OMIM (https://www.omim.org/, accessed on 9 July 2025), have provide structured frameworks to support systematic gene prioritization across CDG subtypes.

Together, these bioinformatics tools and databases form the computational foundation of clinical glycomics, enabling the interpretation of complex glycan datasets, guiding diagnostic strategies, and accelerating biomarker discovery in CDG. As these platforms continue to evolve, their integration into clinical decision-making tools may further streamline the diagnostic process for rare glycosylation disorders.

## 8. Future Directions and Perspectives

The field of clinical glycomics and glycoproteomics in CDG is entering a new era marked by rapid technological growth, computational advancements, and a more integrative understanding of disease biology. Artificial intelligence (AI) and machine learning (ML) techniques have shown promise in analyzing complex glycomic and glycoproteomic datasets. These approaches can reveal subtle nonlinear patterns that may not be detectable using conventional statistical methods. In the context of CDG, AI can assist with the classification of glycosylation patterns, prediction of disease subtypes, and prioritization of candidate variants by integrating multidimensional data from glycomics, genomics, metabolomics, and clinical phenotypes. For example, supervised ML models such as random forests and support vector machines can be trained to distinguish between CDG subtypes based on glycan profile inputs. In parallel, unsupervised clustering algorithms have been applied to group patients based on glycan or proteomic similarity, aiding in the discovery of novel phenotypes and the refinement of ambiguous cases.

AI tools are also increasingly used in variant interpretation pipelines to assess pathogenicity scores, rank likely causative genes, and support automated diagnosis. These computational frameworks, when combined with curated glycosylation databases and phenotype ontologies, have the potential to accelerate diagnostic workflows, increase diagnostic yield, and guide personalized treatment strategies for rare glycosylation disorders. Glycoinformatics is evolving into a powerful data science field, offering significant contributions to diagnostics, precision medicine, and translational research in glycosylation disorders [58,59].

Next-generation MS platforms such as DIA are anticipated to substantially improve the resolution, sensitivity, and throughput of glycoproteomic analyses. This approach enables the systematic capture of all ionized peptides within a sample, regardless of their abundance, thus providing more comprehensive and reproducible glycopeptide datasets. DIA-based workflows have already shown promise in clinical proteomics and are expected to enhance biomarker discovery and therapy monitoring in CDG by improving the quantification of site-specific glycosylation changes across larger patient cohorts [48].

DIA-MS platforms are increasingly being explored for the glycopeptide analysis of clinically relevant plasma proteins such as transferrin, alpha-1-antitrypsin, and immunoglobulin G (IgG), which are known to exhibit glycosylation changes in various disorders [60]. These glycoproteins are of interest due to their diagnostic or prognostic value, and DIA-MS offers the potential to quantify their glycosylation patterns with high reproducibility. The robustness and multiplexing capabilities of this approach make it a promising tool for biomarker development and longitudinal monitoring in CDG.

One promising direction is the expansion of glycomic and glycoproteomic profiling to various biological fluids and tissues. While plasma and serum remain the primary matrices, cerebrospinal fluid, urine, and tissue biopsies can provide insight into organ-specific glycosylation patterns. This is particularly valuable for CDG subtypes with dominant neurological, hepatic, or muscular symptoms, where localized biomarkers may guide precision therapy and improve understanding of disease pathophysiology.

Building international glycoprofile reference libraries and adopting harmonized annotation systems will be instrumental in facilitating global research collaboration and comparative studies. Standardized databases and consistent terminology will help overcome the current barriers in data sharing, allowing researchers to better interpret glycomic profiles across populations and disease subtypes. This infrastructure is essential to support the clinical utility of glycomics and to promote its integration into routine diagnostic workflows.

Glycoproteomic databases are critical for interpreting complex glycosylation patterns and supporting biomarker discovery in CDG. While standard cell-line-based datasets offer baseline information on glycan structures and glycosylation sites, they may not fully capture the tissue-specific or disease-specific glycosylation patterns observed in patients. In contrast, glycoproteomic data derived from patient samples such as serum, fibroblasts, or cerebrospinal fluid provide more relevant insights into the physiological context of CDG subtypes. These patient-derived datasets reflect in vivo variability and disease-related alterations, enabling more accurate diagnostic interpretation and supporting clinical correlation. As glycoproteomics becomes more integrated into translational research, the availability and curation of well-annotated patient-based glycoproteomic datasets will be essential for advancing diagnostics and therapy monitoring in CDG.

Importantly, the integration of glycomics with genomics, transcriptomics, proteomics, and clinical phenotyping is paving the way for systems-level diagnostics. These systems-level approaches enhance diagnostic resolution in genetically unresolved cases by linking genetic variations to their biochemical and functional consequences. For example, transcriptomic data can be used to infer glycosylation capacity in specific tissues. When combined with N-glycomic profiles, these approaches can explain phenotypic variability and help uncover the functional consequences of genetic variants. This level of integration supports the identification of novel disease genes, clarifies the biochemical basis of disease, and enhances diagnostic yield, particularly in genetically unresolved cases.

A novel and emerging approach is the use of stable isotope-labeled sugars to study nucleotide sugar metabolism. This technique allows researchers to functionally assess the biosynthetic pathways involved in glycosylation. In CDG, it can be used to pinpoint metabolic bottlenecks and evaluate the efficiency of therapeutic interventions. For example, isotopic tracing in pluripotent stem cell models has successfully mapped sugar nucleotide biosynthesis, offering valuable information about upstream defects in glycosylation pathways [29].

The development of site-specific glycopeptide biomarkers is another area showing great promise. These markers provide detailed molecular information regarding the glycosylation changes at specific protein sites and may be used to monitor disease activity or therapeutic response. For instance, complement C4-derived glycopeptides have been proposed as specific markers for PMM2-CDG, and transferrin glycopeptides are currently used to assess treatment outcomes in PGM1-CDG. Expanding this strategy to other CDG subtypes could lead to highly personalized monitoring tools for patient management [43,47].

To ensure broad clinical adoption, there is an urgent need for standardization across analytical protocols. This includes harmonizing sample preparation methods, MS acquisition parameters, and data processing workflows. The development of validated bioinformatics pipelines and interoperable databases is equally important for improving reproducibility and enabling inter-laboratory comparisons.

Looking ahead, as personalized medicine continues to evolve, clinical glycoproteomics is likely to become a fundamental tool in individualized patient care. Its capability to identify and quantify site-specific glycan changes across multiple tissues and biofluids opens new diagnostic avenues, especially in neurologically involved or tissue-specific CDG subtypes. Future clinical workflows may include newborn screening programs that utilize glycan-based biomarkers, patient-specific glycosylation profiles for therapeutic decision-making, and dynamic glycoproteomic indices to monitor treatment efficacy over time. These future applications align with the goals of precision medicine and early intervention. These advancements will help shift CDG management from a reactive model to a proactive, precision-guided approach. This evolution prioritizes early detection, targeted intervention, and improved quality of life for affected individuals.

## 9. Conclusions

Clinical glycomics and glycoproteomics have matured into indispensable tools for the diagnosis, subclassification, and monitoring of CDG. This review highlights how advancements in MS, informatics, and multi-omics integration have expanded the clinical utility of glycan-based biomarkers. Through the analysis of transferrin glycoforms, the total plasma N-glycome, site-specific glycopeptides, and emerging biosynthetic and functional readouts, glycomics now offers a multifaceted approach to understanding and managing CDG.

While challenges in standardization, accessibility, and bioinformatics integration remain, the field is moving rapidly toward solutions supported by international guidelines, collaborative databases, and automated platforms. The use of DBS testing, transferrin-based treatment indices, and isotopic tracing exemplifies the translational impact of glycomics in both diagnosis and therapeutic monitoring.

Looking ahead, glycomics is poised to play a central role in precision medicine frameworks. From newborn screening to individualized therapy monitoring, future workflows will integrate glycoprofiling alongside genomic and clinical data to optimize patient care. Glycomic profiles in newborns may exhibit developmental variability compared to older individuals, which can introduce additional complexity in interpretation. Although DBS testing offers a minimally invasive tool for early CDG screening, accurate diagnosis in neonates requires age-specific reference ranges and appropriate validation of glycan-based markers to account for physiological differences at birth. Continued investment in harmonized standards, clinician education, and global research collaboration will be essential to realize the full diagnostic and therapeutic potential of glycomics and glycoproteomics in CDG.

In summary, the convergence of technological innovation, systems biology, and clinical translation marks a promising future for glycomics in rare disease diagnostics. The field is now approaching broader clinical adoption, which will transform how we diagnose, monitor, and treat CDG.

## Figures and Tables

**Figure 1 biomedicines-13-01964-f001:**
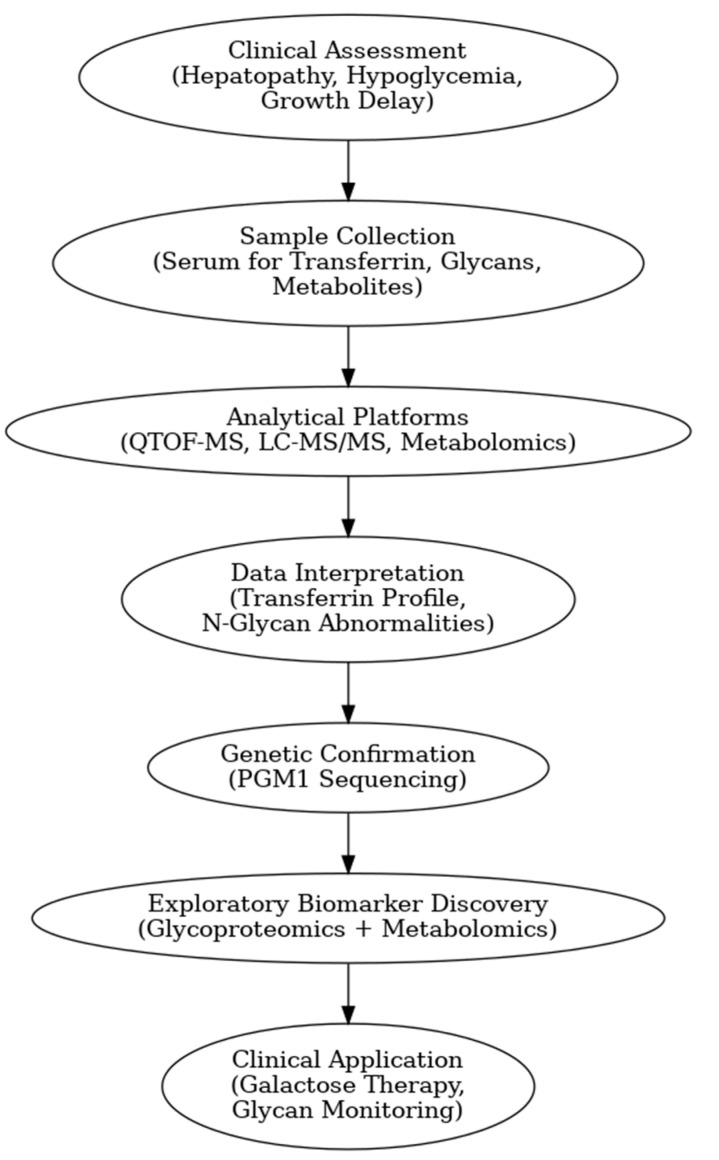
Workflow of clinical and exploratory multi-omics approaches for PGM1-CDG diagnosis and biomarker discovery.

**Table 1 biomedicines-13-01964-t001:** Glycomics-derived biomarkers (glycomarkers) in PGM1-CDG, SLC10A7-CDG, ALG1-CDG, and PMM2-CDG.

CDG Subtype	Glycomarkers Identified	Reference(s)
PGM1-CDG	Total serum or plasma N-glyoprofiling:Increase in total degalactosylated N-glycans.e.g., 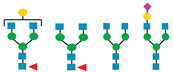 Increase in total fucosylated N-glycans.e.g., 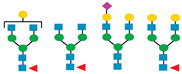 Decrease in total sialylated N-glycans.e.g., 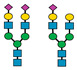	[8,38]
2.Intact transferrin N-glycoprofiling:Increase in three transferrin N-glycans (absence of one complete glycan and galactose residues) for diagnostics. 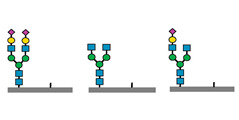 Lack of galactose index monitoring from six transferrin N-glycans during galactose therapy. 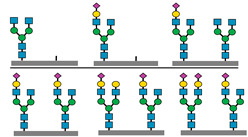 Complete glycan index monitoring from two transferrin N-glycans (absence of one or two complete glycans) during galactose therapy. 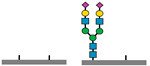 Normal glycosylation index monitoring from the most abundant transferrin N-glycans during galactose therapy. 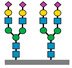
SLC10-CDG	Total serum or plasma N-glyoprofiling:Increase in two total truncated (absence of N-acetylglucosamine residues) N-glycans. 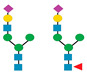 Intact transferrin N-glycoprofiling:ii.Increase in three total truncated (absence of N-acetylglucosamine and sialic acid residues) and one hybrid transferrin N-glycans. 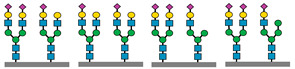	[36]
PMM2-CDG	Total serum or plasma N-glyoprofiling:Mild increase in total N-tetrasaccharide glycan. 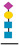 Increase in small total high-mannose N-glycans, especially the 3 mannose residues. 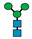 Intact Transferrin N-glycoprofiling:Increase in two transferrin N-glycans (absence of one or two complete glycans). 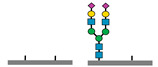 Mild increase in transferrin N-tetrasaccharide glycan. 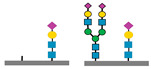	[37,39]
ALG1-CDG	Total serum or plasma N-glyoprofiling:Increase in total N-tetrasaccharide glycan. 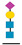 Increase in total fucosylated N-tetrasaccharide glycan. 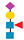 Intact transferrin N-glycoprofiling:Increase in two transferrin N-glycans (absence of one or two complete glycans). 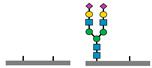 Increase in transferrin N-tetrasaccharide glycan. 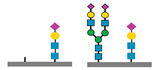	[37,39,40]

Symbol Nomenclature for Glycans: 
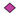
 sialic acids, 
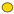
 galactose, 
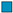
 N-acetyl glucosamine, 
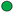
 mannose.

**Table 2 biomedicines-13-01964-t002:** Comparative overview of selected CDG subtypes.

CDG Subtype	Key Glycomarkers	Clinical Features	Diagnostic Tools
PGM1-CDG	↓ Sialylation, ↑ fucosylation, abnormal transferrin N-glycoforms	Hepatopathy, hypoglycemia, myopathy, endocrine abnormalities	Transferrin glycoprofiling, total N-glycome, WES
PMM2-CDG	↓ N-Tetrasaccharide, ↑ high mannose glycans	Developmental delay, hypotonia, ataxia, multiorgan involvement	Transferrin glycoprofiling, total N-glycome, WES
ALG1-CDG	N-Tetrasaccharide, fucosylated glycans	Seizures, developmental delay, muscular hypotonia	Transferrin glycoprofiling, total N-glycome, Sanger sequencing
SRD5A3-CDG	Abnormal N-glycan and glycopeptide profiles	Ocular anomalies, intellectual disability, cerebellar atrophy	Glycoproteomics (fibroblasts), WES
MOGS-CDG	Diagnostic Glc_3_Man tetrasaccharide	Dysmorphic features, developmental delay, feeding problems	Glycan tetrasaccharide screening, enzymatic assay
SLC10A7-CDG	Abnormal N-glycans and abnormal transferrin N-glycoforms	Skeletal dysplasia, dental anomalies, hypotonia	Transferrin glycoprofiling, total N-glycome, WES

## Data Availability

Not applicable.

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
