# Peer review of "Advancement in Clinical Glycomics and Glycoproteomics for Congenital Disorders of Glycosylation: Progress and Challenges Ahead"

_biomedicines, 2025, doi:10.3390/biomedicines13081964_

Round 1
Reviewer 1 Report
Comments and Suggestions for Authors
This review article by Nurulamin and Nurul, titled "Advancements in Clinical Glycomics and Glycoproteomics for Congenital Disorders of Glycosylation: Progress and Challenges Ahead," summarizes the glycan profiles associated with PGM1-CDG, PMM2-CDG, ALG1-CDG, SRD5A3-CDG, MOGS-CDG, and SLC10A7-CDG. The authors aim to provide an update on glycoproteomic technologies and list several glycoproteins that could serve as biomarkers for diagnosis and therapy monitoring.
Overall, the authors briefly touch on well-known concepts but lack in-depth insights or prognoses regarding technological advancements, CDG pathogenesis, or treatment development stemming from glycoproteomic analysis.
Specific comments:
- The authors should introduce dried blood spot testing, isotopic tracing, and combined genetic testing to make the review accessible to general readers.
- It is recommended to include a table comparing the characteristics of PGM1-CDG, PMM2-CDG, ALG1-CDG, SRD5A3-CDG, MOGS-CDG, and SLC10A7-CDG, providing an overview of different CDG deficiencies for broader readers.
- Please enhance the visual appeal of Figure 1 and integrate the information about PGM1-CDG, PMM2-CDG, ALG1-CDG, and SRD5A3-CDG within it.
- In lines 119–120, the author should cite more precise references supporting the use of multi-omics integration for identifying disease mechanisms. For example, some CDGs cannot be easily detected by MS analysis alone and may require transcriptomic or epigenetic insights, highlighting the value of multi-omic approaches.
- There is a duplication of section 7 in the numbering.
- The section on the standardization of diagnostic practices provides limited information; it should be expanded.
- Between lines 324 and 334, the author offers a general summary rather than insights. Specifically, how should genes be prioritized for investigation in CDG diagnosis?
- Between lines 336 and 345, descriptions of how AI can aid CDG data analysis are lacking. Listing specific AI techniques or applications relevant to CDG would strengthen this section.
- From lines 346 to 352, the author should include examples of proteins for which DIA-MS provides glycopeptide information, aiding biomarker identification or treatment monitoring.
- In lines 359 to 365, it would be helpful to discuss what types of glycoproteomic databases (e.g., patient data versus standard cell line databases) are most beneficial for CDG research, along with more detailed insights.
- In Table 1, alongside glycomarkers, existing glycoproteins or peptides mentioned elsewhere in the text should also be annotated.
- Regarding line 417, since glycomic profiles may differ between neonatal and adult blood, the author should comment on whether it is easier or more challenging to detect CDG in newborn blood samples.
Author Response
REVIEWER 1:
This review article by Nurulamin and Nurul, titled "Advancements in Clinical Glycomics and Glycoproteomics for Congenital Disorders of Glycosylation: Progress and Challenges Ahead," summarizes the glycan profiles associated with PGM1-CDG, PMM2-CDG, ALG1-CDG, SRD5A3-CDG, MOGS-CDG, and SLC10A7-CDG. The authors aim to provide an update on glycoproteomic technologies and list several glycoproteins that could serve as biomarkers for diagnosis and therapy monitoring.
Overall, the authors briefly touch on well-known concepts but lack in-depth insights or prognoses regarding technological advancements, CDG pathogenesis, or treatment development stemming from glycoproteomic analysis.
Comments 1: The authors should introduce dried blood spot testing, isotopic tracing, and combined genetic testing to make the review accessible to general readers.
Response 1: Thank you for this thoughtful recommendation. We agree that these diagnostic approaches are important components of the current clinical workflow for CDG. In response, we have introduced dried blood spot (DBS) testing and combined genetic testing in the Introduction to improve accessibility for general readers and to highlight their relevance in routine diagnostic practices.
“In addition, dried blood spot (DBS) testing has emerged as a practical, cost-effective, and minimally invasive approach for sample collection and transport, offering significant advantages for early diagnosis and newborn screening in rare genetic disorders [4].”
(page 2, line 53-56).
New reference: [4]; Malsagova et al. (2020).
“Combined genetic testing, such as the use of next generation sequencing technology alongside glycomic profiling, enhances diagnostic precision and facilitates resolution of uncertain genetic variants.”
(page 2, line 57-59).
Meanwhile, isotopic tracing, which is more specialized and mechanistic in nature, has been elaborated in Section 3 (Integration with multi-omics), where its utility in tracking sugar nucleotide metabolism and supporting variant interpretation is more appropriately discussed.
“Isotopic tracing has gained attention as a mechanistic tool for investigating glycosylation pathways. By following the incorporation of labeled sugars into nucleotide sugar pools and glycan structures, this method provides valuable insights into functional glycosylation defects and supports the biochemical validation of candidate variants. In the context of CDG, isotopic tracing offers a promising approach to elucidate metabolic bottlenecks and assess the efficiency of glycosylation in cellular systems”
(page 4, line 142-148).
All additions have been highlighted in red in the revised manuscript.
Comments 2: It is recommended to include a table comparing the characteristics of PGM1-CDG, PMM2-CDG, ALG1-CDG, SRD5A3-CDG, MOGS-CDG, and SLC10A7-CDG, providing an overview of different CDG deficiencies for broader readers.
Response 2: Thank you for this constructive suggestion. We agree that a comparative overview of selected CDG subtypes would improve clarity and accessibility for readers. In response, we have added a new Table 2, which summarizes the key characteristics of PGM1-CDG, PMM2-CDG, ALG1-CDG, SRD5A3-CDG, MOGS-CDG, and SLC10A7-CDG.
This table includes: Diagnostic glycomarkers, notable clinical features and common diagnostic tools.
(page 10, line 323).
We believe this table complements the glycomics-focused discussion by offering a concise comparison of subtypes referenced throughout the manuscript. Table 2 has been placed in Section 5 (Standardization of Diagnostic Practices) to align with the section’s emphasis on harmonizing diagnostic approaches across clinically diverse CDG presentations.
All additions have been highlighted in red in the revised manuscript.
Comments 3: Please enhance the visual appeal of Figure 1 and integrate the information about PGM1-CDG, PMM2-CDG, ALG1-CDG, and SRD5A3-CDG within it.
Response 3: Thank you for this helpful suggestion. We respectfully clarify that Figure 1 was intentionally designed as a focused, case-based diagnostic workflow centered on PGM1-CDG, a well-characterized subtype with established clinical, biochemical, and therapeutic monitoring strategies. The purpose of this figure is to illustrate the practical application of integrated clinical glycomics, glycoproteomics, and metabolomics in a single representative case, highlighting how multi-omics approaches are implemented in real-world diagnostic settings.
We believe that incorporating additional subtypes into this workflow may reduce visual clarity and compromise the figure’s narrative focus. To address the reviewer’s suggestion for broader representation, we have instead added a new Table 2 that summarizes key characteristics of PMM2-CDG, ALG1-CDG, SRD5A3-CDG, and other relevant subtypes. This table complements Figure 1 by providing a structured comparison across multiple CDGs without compromising the clarity of the visual workflow. We hope this solution provides both illustrative focus and the broader comparative context the reviewer recommended.
Comments 4: The author should cite more precise references supporting the use of multi-omics integration for identifying disease mechanisms. For example, some CDGs cannot be easily detected by MS analysis alone and may require transcriptomic or epigenetic insights, highlighting the value of multi-omic approaches.
Response 4: Thank you for this important suggestion. Additionally, we now provide a specific example involving SRD5A3-CDG in Section 3, where glycomics alone may not sufficiently distinguish the subtype. We explain how integration with transcriptomics and glycoproteomics helps clarify pathogenic mechanisms in such cases.
"For example, in SRD5A3-CDG, glycomics alone may yield overlapping or inconclusive glycan profiles. In such cases, integration with transcriptomics or glycoproteomics can clarify the impact of pathogenic variants by linking genotype to functional glycosylation defects [24]."
(page 3, line 128-137).
We also agree that more specific references and concrete examples would strengthen our discussion of multi-omics integration in CDG diagnostics. In response, we have revised Section 3 to include two additional references:
i. Alvarez et al. [25], which demonstrates how RNA-sequencing-based transcriptomics combined with N-glycomics can predict glycan biosynthesis and tissue-specific expression, aiding in gene prioritization and interpretation of ambiguous variants.
"Alvarez and colleagues demonstrated that combining RNA-sequencing-based transcriptomics with N-glycomics enables the prediction of glycan biosynthesis and tissue-specific glycosylation patterns [25]."
(page 3, line 131-133).
ii. Van der Burgt and Wuhrer [26], which highlights the importance of multi-omics frameworks; particularly glycoproteomics integrated with genomics, in improving diagnostic resolution in rare glycosylation disorders.
"This strategy helps prioritize disease-related genes and interpret ambiguous variant effects. In addition, Van der Burgt and Wuhrer emphasized the role of multi-omics frameworks in precision diagnostics, underscoring how glycoproteomics integrated with genomics enhances diagnostic yield in rare glycosylation disorders [26]."
(page 3, line 133-137).
All additions have been highlighted in red in the revised manuscript and are intended to reinforce the clinical and mechanistic relevance of multi-omics approaches in CDG.
Comments 5: There is a duplication of section 7 in the numbering.
Response 5: Thank you for pointing this out. We have carefully reviewed the manuscript and corrected the section numbering to ensure consistency throughout the document. The duplicate labeling of Section 7 (change to Section 8) has been resolved, and the correct sequential structure has been restored in the revised version. This correction has been highlighted in red in the revised manuscript.
"8. Future directions and perspectives"
(page 12, line 408).
Comments 6: The section on the standardization of diagnostic practices provides limited information; it should be expanded.
Response 6: Thank you for this valuable suggestion. We agree that expanding the section on diagnostic standardization enhances the manuscript’s depth and relevance. In response, we have revised Section 5 to provide a broader and more practical overview of current efforts and tools supporting harmonized CDG diagnostics, especially the role of established clinical glycomics workflows.
Key additions include:
1. We have revised the paragraph to emphasize the role of validated laboratory protocols and scalable mass spectrometry (MS) platforms, including data-independent acquisition (DIA), in standardizing CDG diagnostics and therapy monitoring.
"Recent technical standards by the ACMG have emphasized the need for validated laboratory protocols in CDG diagnostics. These protocols include quality-assured transferrin glycoform analysis, result interpretation, and standardized reporting aimed at harmonizing laboratory practices [16]. As new therapies emerge, MS-based platforms, including data-independent acquisition (DIA) methods, offer scalable and reproducible solutions for analyzing glycosylation changes over time. These tools support consistent measurements across patient cohorts and enable high-throughput monitoring of therapeutic response [48]."
(page 9, line 280-286).
2. A new paragraph emphasizing the standardization of mass spectrometry workflows for clinical glycomics, including established protocols for sample preparation, MS platforms (e.g., MALDI TOF, PGC-LC-MS), data interpretation, and the role of curated MS databases in supporting reproducibility.
“Standardization of mass spectrometry workflows is essential to ensure consistent and accurate glycan analysis in clinical diagnostics. Validated protocols for sample preparation, instrument settings, data acquisition, and interpretation help reduce variability between laboratories. In clinical glycomics, established methods such as MALDI TOF for transferrin profiling and PGC-LC-MS for total plasma N glycome analysis have contributed to improved diagnostic reliability. These approaches rely on clearly defined performance criteria, including detection thresholds, retention time consistency, and reproducible identification of glycan structures. Ongoing collaboration through external quality assessment programs and interlaboratory comparisons continues to strengthen the analytical robustness of glycomics workflows and facilitates their integration into clinical diagnostic pathways. In addition, curated MS databases containing reference glycan spectra support standardized interpretation and facilitate cross-laboratory comparisons. These platforms improve diagnostic consistency and promote reproducibility in glycomics-based testing."
(page 9, line 291-304).
3. Integration of international initiatives, such as ERNDIM proficiency testing (new reference added [49]) and the European Reference Network for Hereditary Metabolic Disorders (MetabERN) (new reference added [50]), which promote global quality assurance, collaborative diagnostics, and improved access to care.
"Efforts to standardize diagnostic practices for CDG have gained momentum through international collaborations and quality assurance programs. Proficiency testing schemes, such as those coordinated by the European Research Network for Evaluation and Improvement of Screening, Diagnosis and Treatment of Inherited Disorders of Metabolism (ERNDIM), have significantly contributed to improving analytical reliability and harmonizing laboratory performance in diagnosing CDG and related disorders [49]. In parallel, expert consensus statements and diagnostic algorithms are being continuously refined to guide clinical interpretation across heterogeneous CDG presentations."
(page 9, line 305-312).
New reference: [49]; Mathis et al. (2022).
"Networking platforms such as the European Reference Network for Hereditary Metabolic Disorders (MetabERN) serve as effective tools for knowledge dissemination, promoting collaboration among clinical and research communities, and supporting the implementation of standardized diagnostic approaches across countries. These efforts ultimately enhance diagnostic accessibility and aim to improve the quality of life for patients and families affected by CDG [50]."
(page 9-10, line 313-318).
New reference: [50]; Péanne et al. (2018).
4. A new Table 2, summarizing glycomarkers, diagnostic tools, and clinical features for selected CDG subtypes, to support harmonization across heterogeneous presentations.
In addition, Table 2 provides a comparative overview of selected CDG subtypes, summarizing their clinical characteristics, glycomarkers, and diagnostic tools. This table is intended to support harmonization by illustrating the diagnostic variability and common analytical strategies used across representative CDG presentations.
Table 2. Comparative overview of selected CDG subtypes
(page 10, line 319-323).
All additions have been highlighted in red in the revised manuscript, and we hope this expanded section meets the reviewer’s expectations.
Comments 7: Between lines 324 and 334, the author offers a general summary rather than insights. Specifically, how should genes be prioritized for investigation in CDG diagnosis?
Response 7: Thank you for this valuable suggestion. In response, we have expanded the relevant paragraph in Section 7 to provide a clearer explanation of gene prioritization strategies in CDG diagnostics. Specifically, we now describe how transferrin glycosylation patterns (type I vs type II) inform the prioritization of genes involved in early glycan assembly or later glycan processing steps. Additionally, we discuss the use of phenotype-driven filtering, clinical features, and sequencing technologies such as molecular inversion probes and whole exome sequencing, which are often interpreted in parallel with glycomics data.
"In clinical practice, gene prioritization for CDG diagnosis is often guided by the patient’s biochemical glycosylation profile and clinical phenotype. For example, a type I transferrin glycosylation pattern typically indicates defects in the early steps of N-glycan assembly, directing attention toward genes involved in cytosolic and endo-plasmic reticulum–based synthesis and transfer of dolichol-linked oligosaccharides, such as PMM2-CDG and ALG1-CDG. Conversely, a type II pattern, reflecting abnor-mal glycan processing or trafficking, suggests defects in Golgi-associated glycosylation, implicating genes such as COG6-CDG or B4GALT1-CDG."
(page 11, line 386-393).
"Additional gene filtering can be guided by phenotype-driven approaches, where features such as predominant liver involvement, developmental delay, or muscular symptoms suggest specific molecular pathways. Diagnostic gene panels and broader sequencing methods, including molecular inversion probes and whole exome sequencing, are increasingly interpreted in combination with glycomic profiles to improve diagnostic precision."
(page 11, line 394-399).
We have also incorporated examples of publicly available resources, such as Orphanet and OMIM, that support systematic gene prioritization by consolidating gene–phenotype relationships and pathway data. These enhancements aim to provide practical insights aligned with current diagnostic workflows.
"Public resources that consolidate gene–phenotype associations and glycosylation pathway data, such as Orphanet (https://www.orpha.net/en/disease) and OMIM (https://www.omim.org/), provide structured frameworks to support systematic gene prioritization across CDG subtypes."
(page 11-12, line 399-402).
"Together, these bioinformatics tools and databases form the computational foundation of clinical glycomics, enabling the interpretation of complex glycan datasets, guiding diagnostic strategies, and accelerating biomarker discovery in CDG. As these platforms continue to evolve, their integration into clinical decision-making tools may further streamline the diagnostic process for rare glycosylation disorders."
(page 12, line 403-407).
The updated section is now more informative while remaining consistent with the scope and style of the review. All changes have been highlighted in red in the revised manuscript.
Comments 8: Between lines 336 and 345, descriptions of how AI can aid CDG data analysis are lacking. Listing specific AI techniques or applications relevant to CDG would strengthen this section.
Response 8: Thank you for this helpful suggestion. We have expanded the relevant paragraph in Section 8 to include a more detailed explanation of how artificial intelligence (AI) and machine learning (ML) techniques can support CDG data analysis. Specifically, we now describe the application of supervised models such as random forests and support vector machines for glycan pattern classification and subtype prediction, as well as unsupervised clustering approaches for patient stratification. We also highlight how AI is being integrated into variant prioritization pipelines to support diagnostics and personalized treatment planning in CDG. These additions aim to provide clearer insights into the role of AI in this evolving field. All revisions have been highlighted in red in the revised manuscript.
"Artificial intelligence (AI) and machine learning (ML) techniques have shown promise in analyzing complex glycomic and glycoproteomic datasets. These approaches can reveal subtle nonlinear patterns that may not be detectable using conventional statistical methods. In the context of CDG, AI can assist with the classification of glycosylation patterns, prediction of disease subtypes, and prioritization of candidate variants by integrating multidimensional data from glycomics, genomics, metabolomics, and clinical phenotypes. For example, supervised ML models such as random forests and support vector machines can be trained to distinguish between CDG subtypes based on glycan profile inputs. In parallel, unsupervised clustering algorithms have been applied to group patients based on glycan or proteomic similarity, aiding in the discovery of novel phenotypes and refinement of ambiguous cases."
(page 12, line 411-421).
"AI tools are also increasingly used in variant interpretation pipelines to assess pathogenicity scores, rank likely causative genes, and support automated diagnosis. These computational frameworks, when combined with curated glycosylation databases and phenotype ontologies, have the potential to accelerate diagnostic workflows, increase diagnostic yield, and guide personalized treatment strategies for rare glycosylation disorders."
(page 12, line 422-427).
Comments 9: From lines 346 to 352, the author should include examples of proteins for which DIA-MS provides glycopeptide information, aiding biomarker identification or treatment monitoring.
Response 9: Thank you for this constructive suggestion. We have revised the relevant section (Section 8) with additional reference [60] to include examples of clinically relevant plasma glycoproteins; such as transferrin, alpha-1-antitrypsin, and immunoglobulin G (IgG), which are known to exhibit glycosylation changes in glycosylation-related disorders. While specific DIA-MS studies on CDG are still emerging, these proteins are widely recognized for their diagnostic and prognostic relevance. We have also clarified the potential of DIA-MS to reproducibly quantify site-specific glycosylation and highlighted its value for biomarker development and longitudinal monitoring in CDG. These additions enhance the clinical perspective without overstating current evidence. All revisions have been highlighted in red in the updated manuscript.
"DIA-MS platforms are increasingly being explored for glycopeptide analysis of clinically relevant plasma proteins such as transferrin, alpha-1-antitrypsin, and immunoglobulin G (IgG), which are known to exhibit glycosylation changes in various disorders [60]. These glycoproteins are of interest due to their diagnostic or prognostic value, and DIA-MS offers the potential to quantify their glycosylation patterns with high reproducibility. The robustness and multiplexing capabilities of this approach make it a promising tool for biomarker development and longitudinal monitoring in CDG."
(page 12, line 437-443).
New reference: [60]; Van Scherpenzeel et al. (2016).
Comments 10: In lines 359 to 365, it would be helpful to discuss what types of glycoproteomic databases (e.g., patient data versus standard cell line databases) are most beneficial for CDG research, along with more detailed insights.
Response 10: Thank you for this valuable suggestion. In response, we have added a new paragraph in Section 8 to expand on the types of glycoproteomic datasets relevant to CDG research. Specifically, we now describe the distinction between standard cell line–based databases and patient-derived datasets. The latter are emphasized as being more informative for understanding disease-specific glycosylation changes, aiding in diagnostic interpretation and biomarker validation. This addition complements the existing discussion on data standardization and collaborative infrastructure. All revisions have been highlighted in red in the revised manuscript.
"Glycoproteomic databases are critical for interpreting complex glycosylation patterns and supporting biomarker discovery in CDG. While standard cell line–based datasets offer baseline information on glycan structures and glycosylation sites, they may not fully capture the tissue-specific or disease-specific glycosylation patterns observed in patients. In contrast, glycoproteomic data derived from patient samples such as serum, fibroblasts, or cerebrospinal fluid provide more relevant insights into the physiological context of CDG subtypes. These patient-derived datasets reflect in vivo variability and disease-related alterations, enabling more accurate diagnostic interpretation and supporting clinical correlation. As glycoproteomics becomes more integrated into translational research, the availability and curation of well-annotated patient-based glycoproteomic datasets will be essential for advancing diagnostics and therapy monitoring in CDG."
(page 13, line 457-467).
Comments 11: In Table 1, alongside glycomarkers, existing glycoproteins or peptides mentioned elsewhere in the text should also be annotated.
Response 11: We appreciate the reviewer’s attention to detail. However, we respectfully clarify that Table 1 is intentionally focused on glycomics-derived biomarkers (glycomarkers), including total plasma N-glycan profiles and intact transferrin glycoform patterns. Its purpose is to highlight characteristic biochemical glycan signatures associated with specific CDG subtypes, without reference to specific protein context. This focus is consistent with the table’s title and scope.
To avoid conceptual overlap, findings at the glycoprotein level such as transferrin glycopeptides and C4-based markers are described in the main text under Section 4: Biomarker Discovery, where they are discussed in relation to emerging glycoproteomic applications. This separation allows us to maintain a clear distinction between glycomics and glycoproteomics within the review structure. We hope this explanation is acceptable.
Comments 12: Regarding line 417, since glycomic profiles may differ between neonatal and adult blood, the author should comment on whether it is easier or more challenging to detect CDG in newborn blood samples.
Response 12: Thank you for this important point. We have now addressed this in Section 9 by briefly commenting on the diagnostic relevance of neonatal samples. In particular, we acknowledge that while dried blood spot testing enables early and minimally invasive sample collection for CDG screening, glycan profiles in newborns may present developmental variability that can complicate interpretation. Therefore, age-specific reference data and validation of glycan markers in neonatal cohorts are essential to ensure accurate diagnosis. All revisions have been highlighted in red in the revised manuscript.
"Glycomic profiles in newborns may exhibit developmental variability compared to older individuals, which can introduce additional complexity in interpretation. Although DBS testing offers a minimally invasive tool for early CDG screening, accurate diagnosis in neonates requires age-specific reference ranges and appropriate validation of glycan-based markers to account for physiological differences at birth."
(page 14, line 523-527).
Reviewer 2 Report
Comments and Suggestions for Authors
The authors present a review on the advancement of clinical glycomics in CDGs. One of the authors was also the author of a review on the subject which is a much-cited reference point in this topic. This review is written in the footsteps of the previous one with the following advances as its theme. I suggest publishing the manuscript after a small change in the title: I would remove the word glycoproteomics since this technique is only touched upon in the review
Author Response
REVIEWER 2:
Comments 1: The authors present a review on the advancement of clinical glycomics in CDGs. One of the authors was also the author of a review on the subject which is a much-cited reference point in this topic. This review is written in the footsteps of the previous one with the following advances as its theme. I suggest publishing the manuscript after a small change in the title: I would remove the word glycoproteomics since this technique is only touched upon in the review.
Response 1: Thank you very much for your kind comments and thoughtful suggestion. We are honoured that you recognized the continuity with our previous work and appreciate your recommendation regarding the manuscript title.
We respectfully propose to retain the term “glycoproteomics” in the title for the following reasons:
1. Expanded scope compared to our 2018 review:
This updated manuscript includes new content that highlights glycoproteomics as an emerging clinical tool in CDG research. Dedicated sections; particularly Sections 3, 4, and 7, discuss its diagnostic relevance, mechanistic insights, and biomarker applications.
2. Distinct clinical utility:
Glycoproteomics is presented not only as a complement to glycomics but also as a platform for protein-specific glycan analysis and therapy monitoring. For example, Section 3 includes the following statement:
“In recent years, glycoproteomics has emerged as a powerful approach to detect subtle, site-specific glycosylation abnormalities in CDG, facilitating both diagnosis and mechanistic understanding of affected pathways.”
(page 4, line 164-166).
3. Clinical translation and future perspectives:
In Section 4, we emphasize its role in biomarker discovery:
“Building upon traditional glycomic biomarkers, glycoproteomics now allows for more refined, protein- and site-specific insights, essential for resolving biochemical phenotypes in complex CDG presentations.”
(page 8, line 224-226).
And in Section 8, we note:
“Its capability to identify and quantify site-specific glycan changes across multiple tissues and biofluids opens new diagnostic avenues, especially in neurologically involved or tissue-specific CDG subtypes.”
(page 13, line 497-500).
These additions collectively reinforce the growing importance of glycoproteomics in CDG diagnostics and support its inclusion in the title. All updates have been clearly marked in red in the revised manuscript, in accordance with MDPI submission guidelines.
We hope this explanation justifies our decision to retain the current title:
“Advancements in Clinical Glycomics and Glycoproteomics for Congenital Disorders of Glycosylation: Progress and Challenges Ahead.”
Reviewer 3 Report
Comments and Suggestions for Authors
This is an interesting review on the application of Glycomics and Proteomics for the diagnosis of Congenital Disorders of Glycosylation. The paper is well written, and the authors’ choices are clearly justified. The Figure and Table are specific and useful, particularly Table 1.
Nevertheless, although many acronyms were defined at the end of the manuscript, some appear in the manuscript that have not been defined, such as PGM1 (phosphoglucomutase 1?), PMM2 (phosphomannomutase 2?), SLC10A7 and ALG1. This reviewer suggests defining all acronyms used in the text.
Author Response
REVIEWER 3:
Comments 1: This is an interesting review on the application of Glycomics and Proteomics for the diagnosis of Congenital Disorders of Glycosylation. The paper is well written, and the authors’ choices are clearly justified. The Figure and Table are specific and useful, particularly Table 1. Nevertheless, although many acronyms were defined at the end of the manuscript, some appear in the manuscript that have not been defined, such as PGM1 (phosphoglucomutase 1?), PMM2 (phosphomannomutase 2?), SLC10A7 and ALG1. This reviewer suggests defining all acronyms used in the text.
Response 1: Thank you very much for your positive feedback and helpful suggestions. We appreciate your emphasis on ensuring clarity regarding acronyms. In response, we have taken the following actions:
1. Added a clarifying sentence early in the Introduction to explain the gene-based nomenclature convention used for CDG subtypes:
“Each CDG subtype is named after the causative gene, followed by the suffix -CDG, reflecting the underlying molecular defect.” (page 1, line 36-38).
2. Defined all acronyms at their first mention throughout the manuscript, including PMM2-CDG, PGM1-CDG and SLC10A7-CDG.
“For example, the most prevalent type of CDG, phosphomannomutase 2 deficiency, is abbreviated as PMM2-CDG.”
(page 1, line 38-39).
“For instance, in phosphoglucomutase 1 deficiency (PGM1-CDG),”
(page 2, line 64-65).
“The identification of solute carrier family 10, member 7 (SLC10A7) as a regulator.............”
(page 5, line 199).
3. Updated the Abbreviations (page 15) section to ensure consistency and completeness with all gene- and CDG-related acronyms used in the text.
All modifications have been clearly highlighted in red in the revised manuscript, in accordance with MDPI submission guidelines.
We thank you again for your thoughtful comment, which has helped improve the readability and clarity of the manuscript.
Round 2
Reviewer 1 Report
Comments and Suggestions for Authors
The authors have addressed the issues from my previous review comments.